# Multi-objective QSAR prediction of ERα antagonists via SHAP-based interpretation

**Jinhui Cao[1]◉, Yanli Liu◉[1,2]◉***

**1** School of Science, Wuhan University of Science and Technology, Wuhan, China, **2** Hubei Province Key Laboratory of Systems Science in Metallurgical Process (Wuhan University of Science and Technology), Wuhan, China

◉ These authors contributed equally to this work.
* yanlil2008@wust.edu.cn

## Abstract

To achieve a comprehensive evaluation of candidate drugs in terms of both biological activity and ADMET properties, this study proposes a two-stage predictive framework based on Quantitative Structure–Activity Relationship (QSAR) modeling integrated with machine learning techniques, elucidating the quantitative relationships between molecular structure and pharmacological properties. A novel Dual-Filter Feature Selection (DFFS) method integrates statistical analysis and feature importance scores derived from machine learning models. The averaged rankings are used to obtain a robust set of molecular descriptors. In the first stage, 20 key two-dimensional molecular descriptors were selected via DFFS from ERα antagonists. RF, XGBoost, LightGBM, and gcForest—were employed for activity prediction. Experimental results indicated LightGBM achieved the best performance, with MRE of 0.0775. The comparative experiment demonstrates that under the same LightGBM regression framework, DFFS outperformed its individual components—Mutual Information and XGBoost—as well as the high-dimensional features generated by ChemBERTa. In the second stage, based on 40 descriptors selected by DFFS, a stacking model was constructed to perform multitask prediction of ADMET properties, ensuring that high-activity candidate compounds also exhibit favorable profiles in absorption, distribution, metabolism, excretion, and toxicity. The AUC scores for all five ADMET models exceeded 0.95. To elucidate the molecular mechanisms and interpret the model decisions, we applied Phi coefficient analysis to assess inter-property correlations and SHAP analysis to identify key molecular features governing compound activity. Furthermore, molecular docking was performed to evaluate the binding affinity of highly active compounds towards the target protein, thereby providing quantitative validation of the predicted biological activities.

**Data availability statement:** The dataset is publicly available at https://doi.org/10.6084/m9.figshare.30459479.

**Funding:** This work was supported by the Hubei Province Key Laboratory of Systems Science in Metallurgical Process (Wuhan University of Science and Technology, https://www.wust.edu.cn/kxyj/kypt.htm) (Grant No. Y202405), awarded to Yanli Liu. The supporting laboratory had no role in the study design, data collection and analysis, decision to publish, or preparation of the manuscript.

**Competing interests:** The authors have declared that no competing interests exist.

## Introduction

Breast cancer (BC) is a malignant tumor caused by the abnormal and uncontrolled proliferation of mammary cells. It is one of the most prevalent malignancies among women worldwide [1]. The Lancet projects that by 2040, global annual breast cancer incidence will exceed 3 million cases, with annual deaths surpassing 1 million, posing a substantial challenge to global public health [2].

BC progression is closely associated with the estrogen receptor alpha (ERα) [3]. In normal mammary epithelial cells, ERα expression typically remains below 10%, but rises significantly in breast cancer tissues [4]. Therefore, compounds that modulate ERα activity are considered potential anti-breast cancer drug candidates, whose biological activity is often assessed by their half-maximal inhibitory concentration (IC50), where lower IC50 values indicate stronger inhibitory effects. IC50 values are commonly converted to their negative logarithmic form, pIC50, establishing a positive correlation with bioactivity in data analysis [5].

In addition to demonstrating strong anticancer activity, candidate compounds must also meet pharmacokinetic and safety requirements, specifically in terms of Absorption, Distribution, Metabolism, Excretion, and Toxicity (ADMET) [6]. The five key ADMET properties include: intestinal epithelial cell permeability (Caco-2) for evaluating drug absorption capacity [7]; cytochrome P450 enzyme subtype 3A4 (CYP3A4) for assessing metabolic stability [8]; cardiac safety (hERG) for evaluating cardiotoxicity [9]; human oral bioavailability (HOB) for measuring the efficiency of systemic drug absorption [10]; and the micronucleus (MN) test for detecting genotoxicity [11]. These properties jointly determine the in vivo behavior of drug candidates and their feasibility for clinical development.

To improve drug development efficiency, compounds targeting ERα and their associated biological properties are often employed as training data for early-stage predictive modeling. Based on a set of molecular descriptors (independent variables) and corresponding biological properties (dependent variables), Quantitative Structure–Activity Relationship (QSAR) models are constructed to elucidate the quantitative correlations between molecular structure and pharmacological effects [12]. These models provide theoretical guidance for virtual screening of novel candidates and structural optimization of existing compounds. Early QSAR models primarily relied on linear regression techniques, while current approaches have evolved into more robust paradigms based on machine learning and deep learning methodologies [13]. To further validate predictive results and elucidate the binding mechanisms between ligands and target proteins, computational chemistry techniques such as molecular docking are frequently employed [14]. Molecular docking simulates the binding process of ligands and receptors in three-dimensional space, predicting optimal binding sites, conformations, and binding affinities. This approach reveals key interacting residues and underlying mechanisms, thereby providing a structural-level interpretation of molecular activity. Integrating molecular docking with predictive models can substantially enhance the reliability and interpretability of drug property prediction [15].

Significant progress has been made in the application of machine learning (ML) to drug discovery. Gomatam et al. [16] utilized the Synthetic Minority Over-sampling Technique (SMOTE) to balance training data and developed a k-nearest neighbor (KNN)-based model for predicting the activity of poly (ADP-ribose) polymerase-1 (PARP-1), achieving high sensitivity and specificity. Banat et al. [17] employed Random Forest (RF) and Extreme Gradient Boosting (XGBoost) models, in combination with QSAR modeling and a genetic algorithm, to evaluate six groups of Aurora-A kinase (AURKA) inhibitors, identifying three compounds with potent inhibitory activity against AURKA. Similarly, Di Stefano et al. [18] developed a toxicological prediction platform named VenomPred 2.0, which integrates KNN, Support Vector Machine (SVM), RF, and Multilayer Perceptron (MLP) models along with SHapley Additive exPlanations (SHAP) for interpretable predictions across multiple toxicity endpoints of chemical substances. Meanwhile, deep learning approaches have been increasingly applied to enhance feature extraction and model prediction capabilities. Zhang et al. [19] proposed a novel multimodal architecture called ISMol, which validated the complementarity between molecular images and SMILES strings. Jung et al. [20] leveraged the pretrained natural language processing model ChemBERTa, combined with deep neural networks (DNNs), encoders, concatenation layers, and pipelines, for QSAR modeling of drug ADMET profiles. They emphasized the necessity of larger and more diverse datasets to improve model generalizability.

To address the limitations of existing models in the comprehensive evaluation of biological activity and ADMET properties, we propose a novel feature selection method—Dual-Filter Feature Selection (DFFS)—which integrates statistical analysis with model-based evaluation to uncover key associations between molecular descriptors and multiple drug-related properties. Based on DFFS, a two-stage predictive framework was established to progressively identify compounds with favorable pharmacological profiles. In the first stage, an activity prediction model was developed using data from ERα antagonists to enable preliminary screening of highly active compounds. In the second stage, the top-ranked active candidates were further evaluated across five ADMET properties, and a Stacking ensemble model was employed to enhance predictive performance. Ablation studies combined with Phi coefficient analysis, SHAP interpretation, and molecular docking were conducted to interpret model decisions and elucidate underlying molecular mechanisms, thereby providing mechanistic insights into compound–target interactions.

## Feature engineering

The dataset [21], sourced from the ChEMBL database, contains 1,974 compounds with labeled activity and ADMET properties for model training, and 50 compounds with unlabeled data for independent validation. All molecular structures are represented in the SMILES format. A total of 729 two-dimensional molecular descriptors covering 47 categories were calculated using PaDEL descriptors in ChemDes. During data preprocessing, 225 descriptors with constant zero values and no predictive contribution were removed. The remaining features were standardized using min–max normalization to eliminate scale discrepancies. The biological activity of annotated compounds was treated as a continuous variable, while the five ADMET properties were encoded as binary classification variables, where 1 indicates favorable performance and 0 denotes poor performance or the absence of that property.

To obtain a robust and representative set of molecular features, we propose a dual-filter feature selection method that integrates both statistical analysis and machine learning-based evaluations. Specifically, feature importance was assessed using Mutual Information (MI), RF, and XGBoost, and the results were aggregated to mitigate the limitations of any single method. MI, as a statistical measure quantifying the dependency between molecular descriptors and compound properties, evaluates the degree of shared information by comparing the joint probability distribution $p(x)$ and $p(y)$. The MI value reflects the strength of the relationship: the greater the MI, the stronger the correlation between variables; an MI of zero indicates statistical independence. This relationship is mathematically described in Equation (1) [22].

$$I(X; Y) = \sum_{x \in X} \sum_{y \in Y} p(x, y) \log \left( \frac{p(x, y)}{p(x)p(y)} \right)$$

(1)

The RF and XGBoost models adopt the Bagging and Boosting paradigms, respectively, by training multiple base learners to capture the nonlinear relationships in the data [23]. Feature importance in RF is evaluated based on the frequency of feature usage during node splits and the decrease in Gini impurity. Features that are frequently used for splitting and contribute significantly to impurity reduction are assigned higher importance scores, indicating a greater impact on the target prediction. In contrast, XGBoost evaluates feature importance using metrics such as information gain, coverage, or frequency of usage. Within the DFFS framework, the cumulative information gain of each feature during decision tree construction is computed and the features are ranked in descending order of importance.

For each feature, its position in the ranked importance lists derived from the three methods is denoted as $rank_{MI}$, $rank_{RF}$, and $rank_{XGBoost}$, respectively. The average rank is then calculated using Equation (2), where a lower average rank indicates higher overall importance.

$$average\ rank = \frac{rank_{MI} + rank_{RF} + rank_{XGBoost}}{3}$$

(2)

Two final feature subsets were selected: $D_{Activity}$, consisting of 20 molecular descriptors for predicting biological activity; and $D_{ADMET}$, consisting of 40 descriptors for predicting each of the five ADMET properties. These descriptors encode a diverse range of physicochemical, structural, and topological information, thereby ensuring a comprehensive representation of compound characteristics. Table 1 summarizes the descriptors included in $D_{Activity}$.

### Two-stage prediction model for anti-breast cancer drug candidates' performance

To reduce time and cost in drug development, activity prediction models are often established to screen potential active compounds. Fig 1 illustrates the two-stage modeling workflow: a regression model is first constructed to predict biological activity, followed by a classification model for ADMET property prediction. Following the proposed DFFS method to obtain the training set for activity ($D_{Activity}$) and ADMET properties ($D_{ADMET}$), the dataset was partitioned into training and testing subsets at an 8:2 ratio. Subsequently, the 80% training set was utilized for model construction, employing a consistent five-fold cross-validation procedure. It is important to note that a compound was included in $D_{ADMET}$ only if its predicted pIC50 value from the activity regression model exceeded 6, thereby guaranteeing that the screened compounds exhibit high activity [24,25]. To further enhance model performance and automate hyperparameter optimization, the Tree-structured Parzen Estimator (TPE) algorithm based on Bayesian optimization was employed for global parameter search and tuning, thereby obtaining the optimal combination of model parameters [26].

**Table 1. Presentation of features for activity screening.**

| Features for predicting pIC50 | Descriptor type |
|---|---|
| ATSc3 | Autocorrelation (charge) |
| BCUTc-1l, BCUTp-1h, BCUTc-1h | BCUT |
| C1SP2 | Carbon types |
| maxHsOH, minHsOH, minsOH, minssO, LipoaffinityIndex, TopoPSA, mindssC, maxsOH, SHBint6, maxHBd | Atom type electrotopological state |
| MLogP | Mannhold LogP |
| MDEO-12 | Molecular distance edge |
| MDEC-23, MLFER_A | Molecular linear free energy relation |
| minHBint5 | Topological polar surface area |

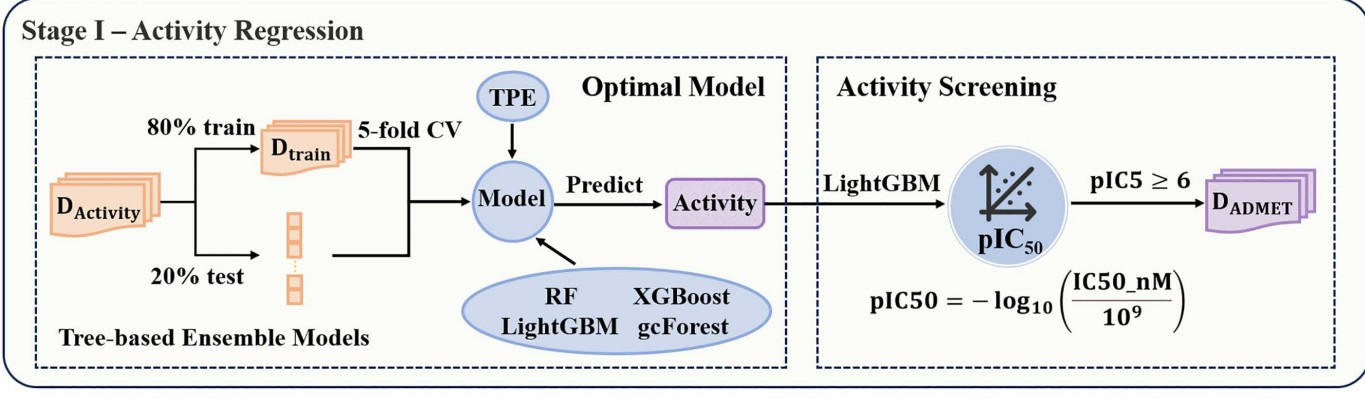

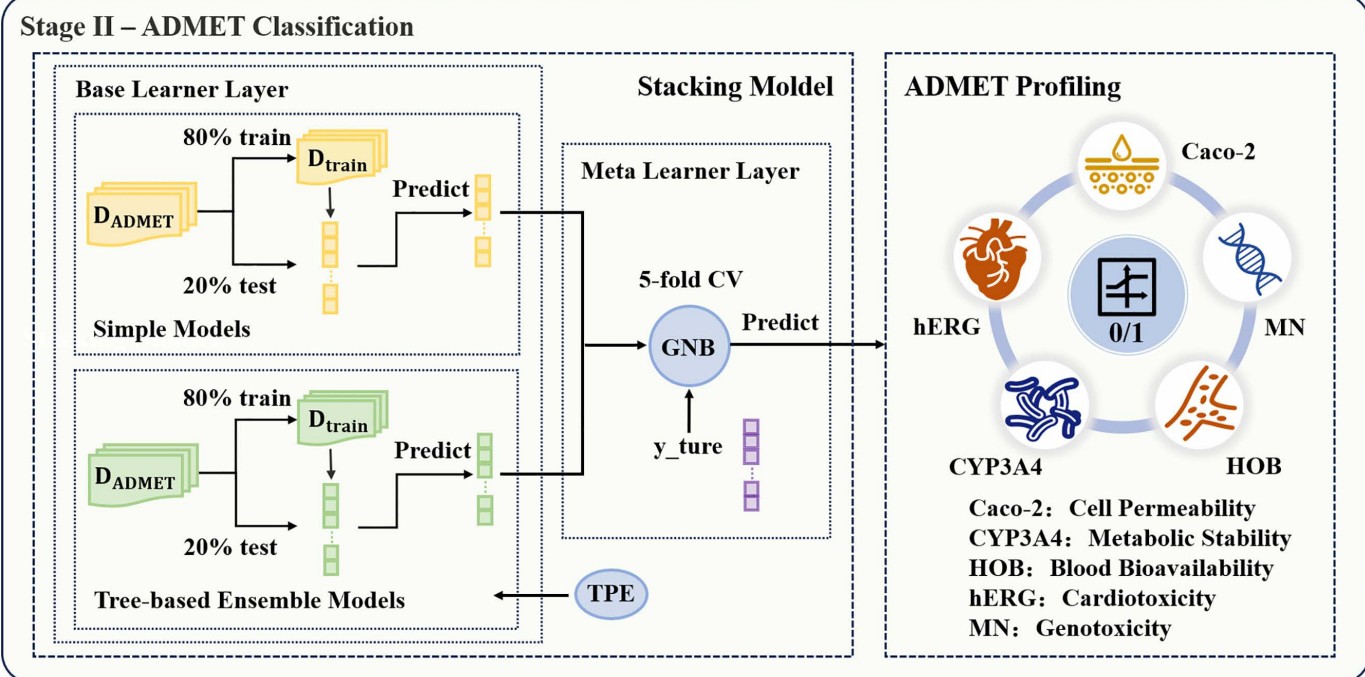

**Fig 1. Two-stage target prediction flowchart.**

## Screening model based on activity prediction

Given the complexity of the $D_{Activity}$ descriptor set, we employed four ensemble learning models—RF, XGBoost, Light Gradient Boosting Machine (LightGBM), and Deep Forest (gcForest)—which are inherently robust to outliers. RF model employs bootstrapping to randomly sample data and selects a subset of features at each split, thereby reducing computational cost for base learners [27]. XGBoost constructs additive models by sequentially fitting residuals and minimizes loss via first- order and second-order derivatives, with a regularization term incorporated to prevent overfitting [28]. LightGBM enhances efficiency by using a histogram-based approach to discretize continuous features and adopts a leaf-wise growth strategy to minimize prediction error. It also introduces techniques such as Gradient-based One-Side Sampling (GOSS) and Exclusive Feature Bundling (EFB) to improve scalability to large datasets [29]. gcForest consists of multi-grained scanning and a cascade forest structure. It generates multi-scale feature representations using sliding windows, and concatenates the prediction results of each RF with the original features, thereby enhancing model performance [30].

The performance of the activity regression prediction model was evaluated using four metrics. The Mean Relative Error (MRE) measures the relative deviation of predicted values from the true values. The Mean Squared Error (MSE) reflects the average squared deviation between predicted and actual values, while the Mean Absolute Error (MAE) represents the average absolute difference between them. The coefficient of determination ($R^2$) assesses the proportion of total variance in the observed data that is explained by the model and serves as an important indicator of the goodness of fit for regression models [31]. Smaller MRE, MSE, and MAE values approaching 0 and an $R^2$ value closer to 1 indicate higher predictive accuracy and better model fitting performance.

$$MRE = \frac{1}{n} \sum_{i=1}^{n} \left( \frac{|y_i - \hat{y}_i|}{|y_i|} \right) \tag{3}$$

$$MSE = \frac{1}{n} \sum_{i=1}^{n} (y_i - \hat{y}_i)^2 \tag{4}$$

$$MAE = \frac{1}{n} \sum_{i=1}^{n} |y_i - \hat{y}_i| \tag{5}$$

$$R^2 = 1 - \frac{\sum_{i=1}^{n} (y_i - \hat{y}_i)^2}{\sum_{i=1}^{n} (y_i - \bar{y})^2} \tag{6}$$

where $n$ is the total number of data samples, $\hat{y}_i$ denotes the predicted value, $y_i$ is the actual value, and $\bar{y}$ represents the mean of the observed values.

As shown in Table 2, LightGBM achieved the best performance across all evaluation metrics, followed by gcForest and XGBoost, while RF exhibited relatively weaker performance. Previous studies have suggested that when the $R^2$ value on the test set reaches or exceeds 0.6, the model can generally be considered to possess reasonable predictive capability that meets the scientific requirements for drug activity prediction [32]. In this study, LightGBM achieved an $R^2$ of 0.7471 and an MRE of 0.0775, indicating that the model exhibited low prediction bias across the overall sample range, with high fitting accuracy and stability. These results demonstrate that LightGBM based on DFFS is well-suited for supporting drug activity prediction and virtual screening applications.

Fig 2 further provides a detailed analysis of the proportion of samples across different intervals of MRE. As the prediction deviation increases, the proportion of samples gradually decreases. Notably, nearly half of the test samples predicted by the LightGBM model fall within the 0–5% error range, highlighting its superior robustness. Furthermore, Fig 2 indicates that all four ensemble learning models exhibit a small number of extreme errors (relative error >30%), suggesting that certain compounds are particularly challenging to predict accurately.

**Table 2. Evaluation metrics of activity prediction models.**

| Models | MRE | MSE | MAE | R² |
|---|---|---|---|---|
| RF | 0.0898 | 0.5921 | 0.5767 | 0.6923 |
| XGBoost | 0.0849 | 0.5352 | 0.5443 | 0.7218 |
| LightGBM | **0.0775** | **0.4865** | **0.4959** | **0.7471** |
| gcForest | 0.0784 | 0.5008 | 0.5046 | 0.7397 |

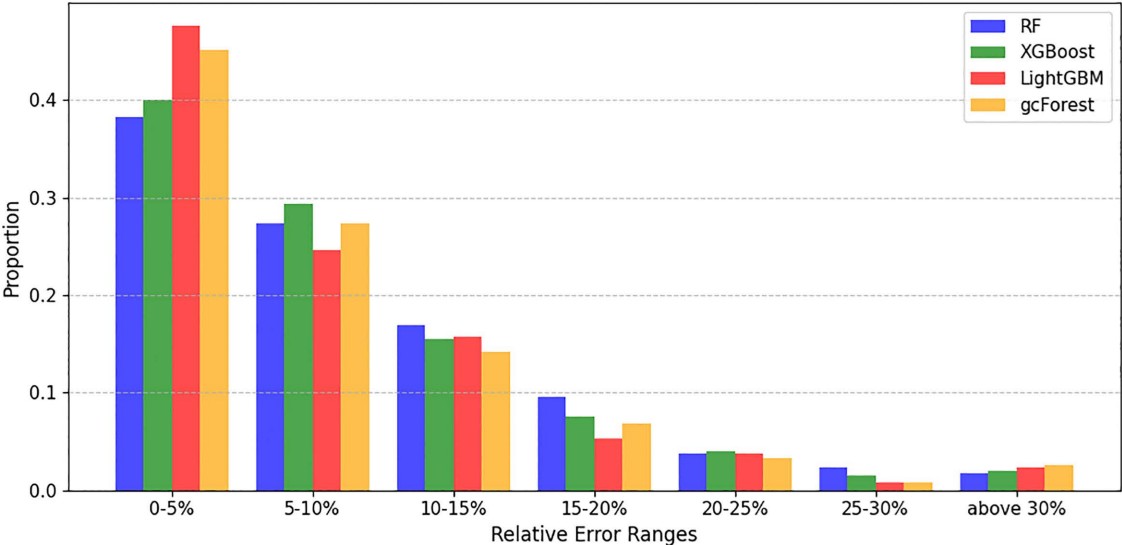

**Fig 2. Histogram of relative error distribution for activity prediction models.**

To validate the effectiveness of the proposed selection method, we compared DDFS with MI, XGBoost, and the state-of-the-art ChemBERTa under the optimal LightGBM regression framework. The ChemBERTa model is deep learning–based to extract features from the SMILES representations of compounds. ChemBERTa, a self-supervised masked language model (MLM), is pretrained to capture the semantic and contextual dependencies embedded in molecular structures, encoding each molecule into a fixed-length 768-dimensional vector [33]. To reduce feature redundancy and enhance computational efficiency, principal component analysis (PCA) was applied for dimensionality reduction [34], compressing the feature space from 768 to 97 dimensions while retaining 95% of the total information content. This ensured a balance between structural integrity and model computability.

Table 3 indicates that DFFS achieved the best performance across all evaluation metrics, demonstrating its superior overall effectiveness in feature selection. In contrast, when MI or XGBoost was used individually for feature selection, model performance decreased slightly, suggesting that single methods are limited in their ability to capture complex feature correlations.

## Screening model based on ADMET prediction

According to the screening criterion of pIC50 ≥ 6, a total of 1,231 highly active candidate compounds were identified. Compared with 743 low-activity compounds, the overall sample distribution was relatively balanced. Based on the $D_{ADMET}$ dataset, Stacking ensemble models were constructed for each ADMET property, including Caco-2, CYP3A4, hERG, HOB, and MN. To fully leverage the complementary strengths of heterogeneous base learners and overcome the limitations of

**Table 3. Comparison of model performance using different feature selection methods.**

| Feature selection method | MRE | MSE | MAE | R² |
|---|---|---|---|---|
| **DFFS-LightGBM** | **0.0775** | **0.4865** | **0.4959** | **0.7471** |
| **MI-LightGBM** | 0.0788 | 0.5026 | 0.5028 | 0.7388 |
| **XGBoost-LightGBM** | 0.0818 | 0.5254 | 0.5226 | 0.7269 |
| **ChemBERTa-LightGBM** | 0.1032 | 0.7045 | 0.6507 | 0.6338 |

**Table 4. Composition of the stacking model in ADMET prediction.**

| Property | Base learner | Meta learner | K-fold CV |
|---|---|---|---|
| Caco-2 | SVM + XGBoost+LightGBM | GNB | 5 |
| CYP3A4 | KNN + RF + XGBoost+LightGBM | GNB | 5 |
| hERG | KNN + RF + XGBoost | GNB | 5 |
| HOB | KNN + XGBoost | GNB | 5 |
| MN | XGBoost+LightGBM | GNB | 5 |

single models, various combinations of SVM, KNN, RF, XGBoost, and LightGBM were explored for each target variable. The optimal base learner combinations for predicting the five properties are summarized in Table 4. A Gaussian Naive Bayes (GNB) model was employed as the meta-learner to effectively integrate the outputs of the base learners and generate the final class predictions.

The evaluation of ADMET classification prediction models was conducted using five key metrics. The Accuracy measures the proportion of compounds with favorable properties (positive class) that are correctly predicted by the model. The Precision quantifies the proportion of truly positive samples among those predicted as positive. The Recall reflects the model's ability to identify positive samples. The F1 score, defined as the harmonic mean of Precision and Recall, is particularly suitable for evaluating performance on imbalanced datasets. The Area Under the Curve (AUC) assesses the model's overall discriminative power by calculating the area under the Receiver Operating Characteristic (ROC) curve, where the ROC plots the True Positive Rate (TPR) on the y-axis against the False Positive Rate (FPR) on the x-axis. Higher metric values closer to 1 indicate better model performance [29].

$$Accuracy = \frac{TP+TN}{TP+TN+FP+FN} \tag{7}$$

$$Precision = \frac{TP}{TP+FP} \tag{8}$$

$$Recall = \ TPR = \frac{TP}{TP+FN} \tag{9}$$

$$F1 = 2 \times \frac{Precision \times Recall}{Precision+Recall} \tag{10}$$

$$FPR = \frac{FP}{TN+FP} \tag{11}$$

where TP (True Positive) is the number of positive samples correctly identified by the model, FP (False Positive) is the number of negative samples incorrectly predicted as positive, TN (True Negative) is the number of negative samples correctly recognized, and FN (False Negative) is the number of positive samples incorrectly predicted as negative.

Fig 3 provides a detailed comparison of the predictive performance for Caco-2, CYP3A4, hERG, HOB, and MN across different evaluation metrics. The data points indicate that the Stacking model proposed in this study consistently outperforms all other candidate models across all five ADMET properties.

Table 5 further compares the performance metrics between the optimal Stacking model and the best-performing individual base learner for each property. The Stacking model demonstrates superior performance, particularly in terms of Accuracy and F1-score, with AUC values consistently exceeding 0.95. This indicates that the model not only achieves high

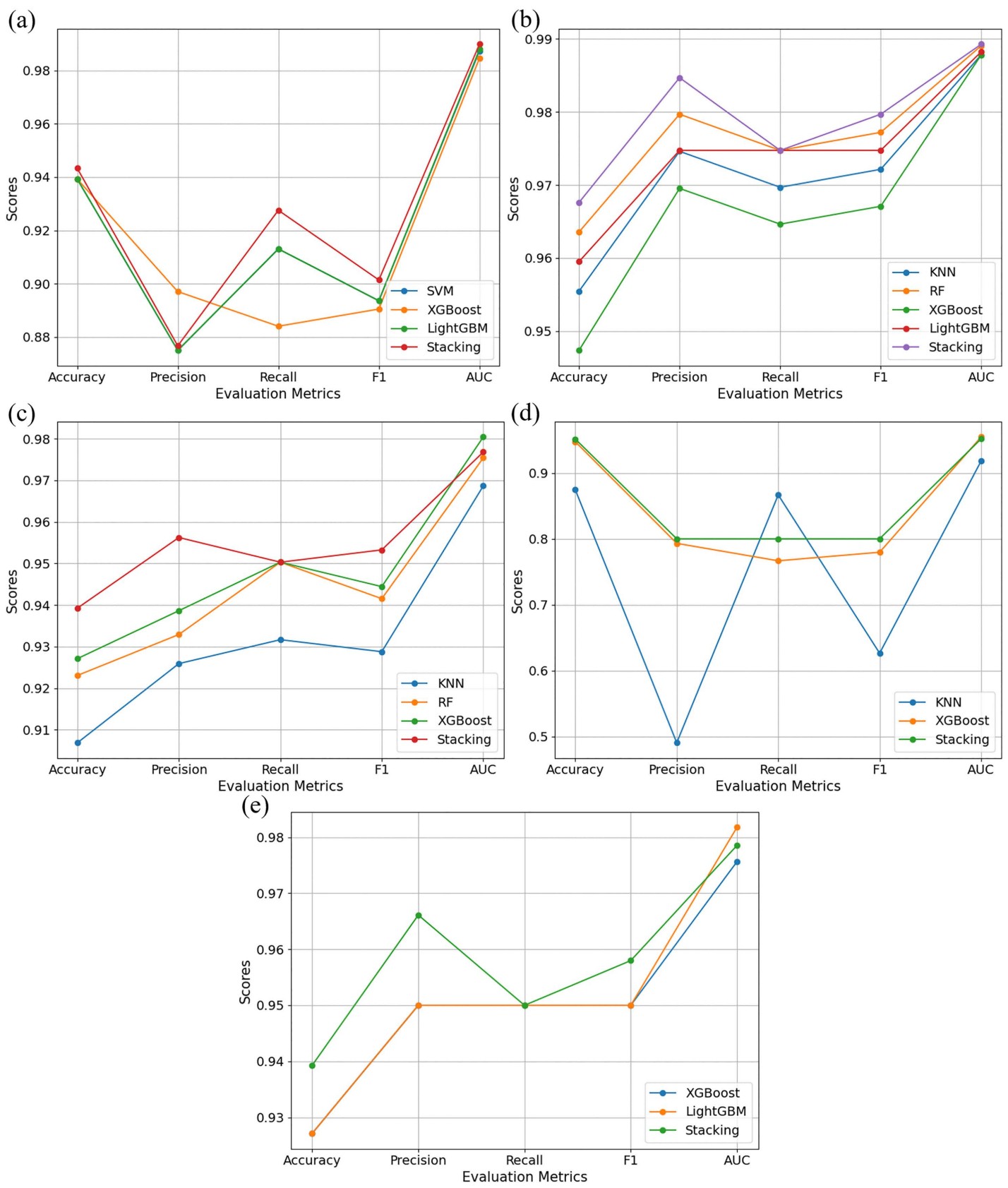

**Fig 3. Comparison of model evaluation metrics for ADMET prediction. (a)–(e) correspond to Caco-2, CYP3A4, hERG, HOB, and MN.**

**Table 5. Evaluation metrics of the stacking model vs. Best single model in ADMET prediction.**

| Property | Model | Accuracy | Precision | Recall | F1 | AUC |
|---|---|---|---|---|---|---|
| Caco-2 | Stacking | **0.9433** | **0.8767** | **0.9275** | **0.9014** | **0.9898** |
| | LightGBM | 0.9393 | 0.8750 | 0.9130 | 0.8936 | 0.9879 |
| CYP3A4 | Stacking | **0.9676** | **0.9847** | **0.9747** | **0.9797** | **0.9893** |
| | RF | 0.9636 | 0.9797 | 0.9747 | 0.9772 | 0.9891 |
| hERG | Stacking | **0.9393** | **0.9563** | **0.9503** | **0.9533** | **0.9768** |
| | XGBoost | 0.9271 | 0.9387 | 0.9503 | 0.9444 | 0.9804 |
| HOB | Stacking | **0.9514** | **0.8000** | **0.8000** | **0.8000** | **0.9518** |
| | XGBoost | 0.9474 | 0.7931 | 0.7667 | 0.7797 | 0.9545 |
| MN | Stacking | **0.9393** | **0.9661** | **0.9500** | **0.9580** | **0.9785** |
| | LightGBM | 0.9271 | 0.9500 | 0.9500 | 0.9500 | 0.9818 |

predictive accuracy but also exhibits excellent robustness in ADMET property prediction. The consistent outperformance of the Stacking model across all evaluation metrics suggests that it effectively integrates the strengths of heterogeneous base learners, resulting in more reliable and precise predictions.

## Model interpretability and molecular mechanism analysis

### Evaluation of correlations among ADMET properties

Pharmacological properties of compounds often exhibit potential interrelationships [35]. To quantify the degree of association among the five ADMET properties of highly active compounds, the Phi coefficient was employed, and the significance of these associations was further assessed using the chi-square test to validate the effectiveness of the Phi coefficient. The Phi coefficient ranges from −1–1, with positive values indicating positive correlation and negative values indicating negative correlation [36].

$$\phi = \frac{n_{11}n_{00} - n_{10}n_{01}}{\sqrt{(n_{1.}n_{0.}n_{.1}n_{.0})}}$$

(12)

where $n_{11}$, $n_{00}$, $n_{10}$, and $n_{01}$ represent the observed frequencies of different combinations of the five ADMET binary properties, and $n_{1.}$, $n_{0.}$, $n_{.1}$, $n_{.0}$ denote the corresponding marginal frequencies.

Based on this dataset, the statistical associations among ADMET properties are summarized in Table 6. A significant negative correlation was observed between Caco-2 and hERG (Phi = −0.4393, p < 0.001), suggesting that good intestinal permeability may contribute to a reduced risk of cardiotoxicity. Caco-2 also exhibited a significant negative correlation with CYP3A4 and a significant positive correlation with HOB, indicating that drug design requires balancing enhanced bioavailability with optimized metabolic properties. In addition, a significant positive correlation was found between hERG and CYP3A4 (Phi = 0.5468, p < 0.001), highlighting the need to consider potential cardiotoxicity risks during drug development. In contrast, MN showed no significant correlations with CYP3A4, hERG, or HOB, suggesting that genotoxicity is relatively independent of these properties in this dataset.

### SHAP analysis of the activity model

The pIC50 directly reflects the inhibitory potency of compounds against ERα, making the analysis of key molecular descriptors that influence activity crucial for improving the efficiency of drug design and synthesis. To quantify the

**Table 6. Statistical analysis of ADMET correlations.**

| Phi coefficient | Caco-2 | CYP3A4 | hERG | HOB | MN |
|---|---|---|---|---|---|
| Caco-2 | 1 | −0.3863*** | −0.4393*** | 0.2866*** | −0.2194*** |
| CYP3A4 | | 1 | 0.5468*** | −0.1839*** | 0.0121 |
| hERG | | | 1 | −0.2349*** | −0.0416 |
| HOB | | | | 1 | 0.0297 |
| MN | | | | | 1 |

*** p < .001.

contribution of each molecular descriptor to the model predictions, SHAP was employed for interpretative analysis. SHAP is a model-agnostic feature importance method based on the Shapley value concept from cooperative game theory, which evaluates the marginal contribution of each feature to the prediction [18]. Compared with traditional feature evaluation methods that rely on internal model metrics, SHAP is applicable to both linear and nonlinear models and can reveal interactions and non-additive effects among features.

Fig 4 presents the SHAP feature importance bar chart and beeswarm plot for pIC50, illustrating the average feature importance and their contribution to the activity prediction. The bar chart on the upper X-axis displays the average absolute SHAP values for each feature, ranked by importance. Among the features, LipoaffinityIndex and MDEC-23 have the most significant impact on pIC50. Specifically, the SHAP values for these two features are 0.2491 and 0.2009, respectively. The beeswarm plot on the lower X-axis shows the distribution of SHAP values for each feature. Positive values indicate a positive impact, while negative values indicate a negative impact. The color gradient indicates the feature value

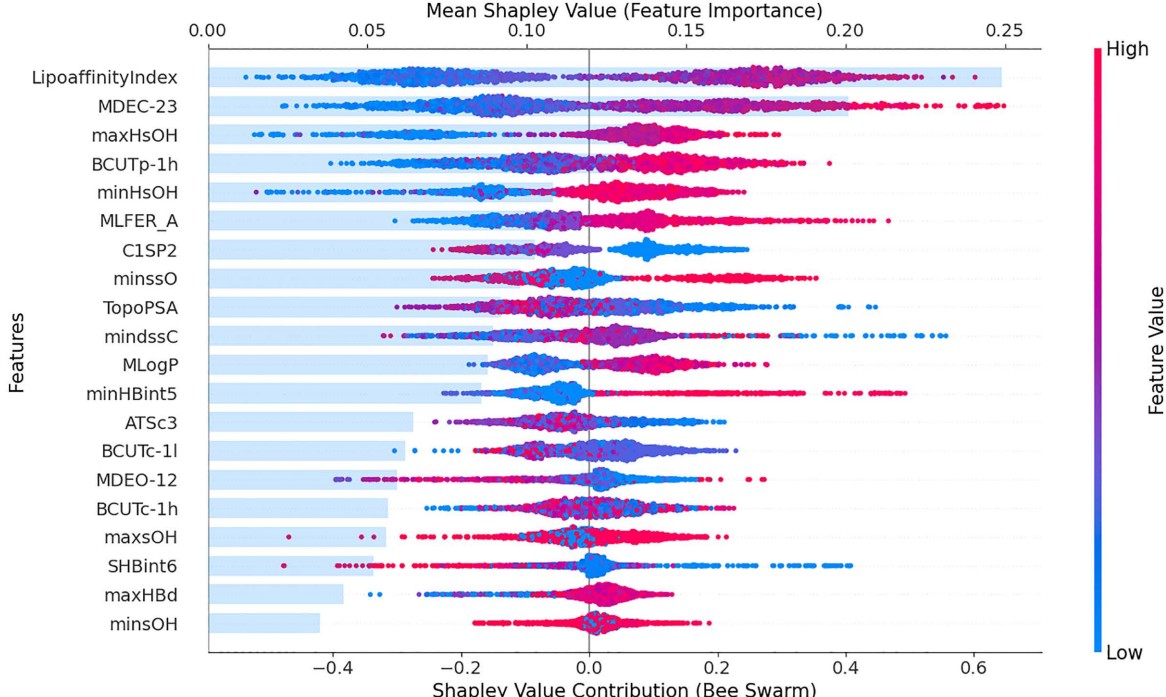

**Fig 4. SHAP global explanation plot for pIC50.**

level (with red for higher and blue for lower values). The beeswarm plot shows that larger values of LipoaffinityIndex and MDEC-23 have a stronger positive effect on pIC50. Fig 5 further examines the numerical distribution of these two descriptors. Each point in the figure represents a molecular sample with its corresponding descriptor value and predicted activity, visualizing the distribution of the values across samples and their potential contributions to pIC50.

Fig 5a shows that when the LipoaffinityIndex exceeds 8, it favors an increase in activity, indicating a pronounced threshold effect for this molecular descriptor. The LipoaffinityIndex reflects the lipophilicity of a compound, representing its affinity for lipid environments. This observation aligns with the mechanism of action of ERα antagonists. As an intracellular nuclear receptor, ERα requires ligands to traverse the cell membrane before binding. Higher lipophilicity facilitates passive diffusion across the lipid bilayer, enabling effective intracellular exposure and thereby enhancing apparent inhibitory activity [37]. Moreover, recent studies have highlighted that the dynamic balance between lipophilicity and receptor binding affinity is critical for optimizing the pharmacological efficacy of selective estrogen receptor modulators (SERMs) [38]. Moderate lipophilicity typically achieves an optimal balance between membrane permeability and receptor binding, leading to improved bioactivity [39].

Fig 5b illustrates that when MDEC-23 exceeds 20, its effect on pIC50 shifts from negative to positive. MDEC-23 represents the mean distance between secondary and tertiary carbon atoms in a molecule, reflecting the spatial extension and flexibility of the molecular scaffold. This trend aligns with the structural characteristics of the ERα binding site [40]. The ligand-binding pocket of ERα is relatively spacious, moderately hydrophobic, and conformationally plastic, allowing dynamic adjustment of ligand conformation via an "induced fit" mechanism [41]. Recent structural biology and structure–activity relationship (SAR) studies confirm that moderate molecular extension and flexibility facilitate conformational adaptation within the binding pocket, thereby enhancing ERα binding affinity [42]. In contrast, lower MDEC-23 values correspond to compact, rigid molecules that, despite higher intrinsic stability, exhibit limited conformational adaptability. Conversely, higher MDEC-23 values indicate greater molecular flexibility and spatial extensibility, enabling the formation of stable complexes with the deformable binding pocket [43].

## Molecular docking and binding mode analysis

Molecular docking systematically explores multiple orientations and conformations of ligands within the receptor's active site and calculates ligand–receptor binding energies using scoring functions to identify the most favorable binding modes

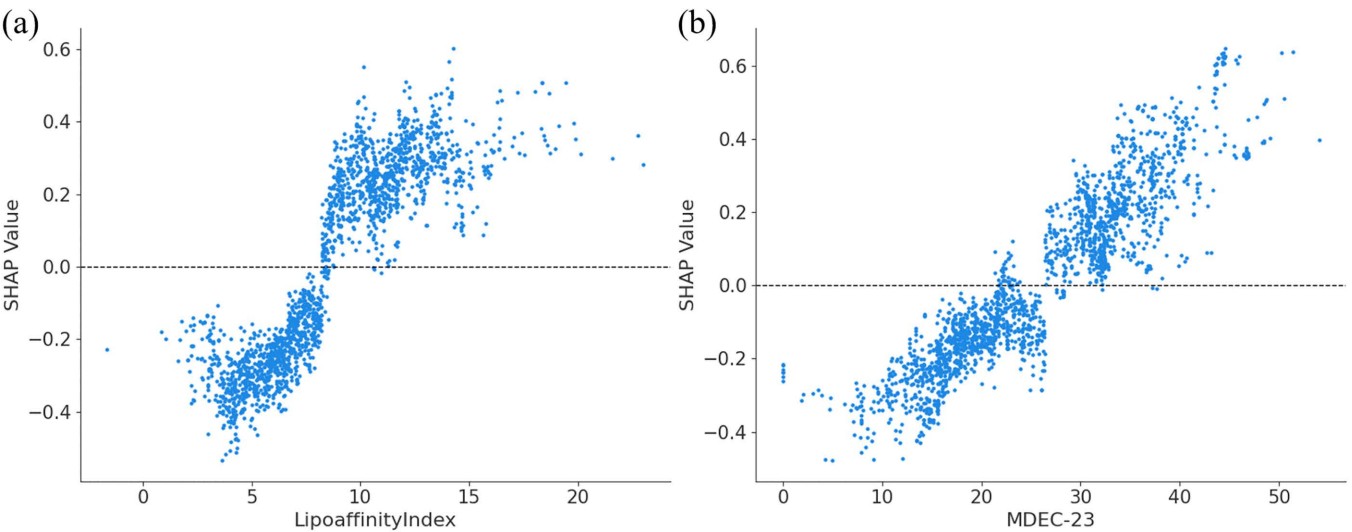

**Fig 5. SHAP scatter plot for pIC50. (a)–(b) correspond to LipoaffinityIndex and MDEC-23.**

[44]. Activity predictions of 50 unlabeled compounds indicated that 14 compounds had predicted pIC50 values greater than 7, 30 compounds were between 6 and 7, and 6 compounds were below 6. From these, the top five compounds ranked by predicted pIC50 (8.0721, 7.5194, 7.4693, 7.4377, and 7.4263) were selected for docking with ERα, and their chemical structures are shown in Fig 6a-Fig 6e. The docking results revealed that the binding free energies (ΔG) of all five compounds were below −7.0 kcal/mol, indicating strong binding affinity and favorable spatial complementarity within the ERα binding pocket, sufficient to form stable complexes under physiological conditions [45]. Notably, the compound exhibiting the lowest binding free energy (ΔG = −8.3 kcal/mol) also displayed the highest predicted activity (pIC50 = 8.0721), indicating a consistent correlation between binding affinity and predicted potency.

Furthermore, we conducted a systematic structural analysis and interaction visualization for the five high-activity compounds in complex with ERα, as shown in Fig 7. Fig 7a-Fig 7d illustrate the binding mode of Compound 1, with the optimal binding conformation exhibiting a ΔG of −8.3 kcal/mol, indicating a stable configuration. The remaining binding modes showed ΔG values ranging from −7.9 to −7.1 kcal/mol, all demonstrating substantial binding activity. Analysis revealed that Compound 1 forms a stable complex with ERα through multiple non-covalent interactions, particularly a hydrogen bond with residue ARG-349, significantly enhancing binding affinity and complex stability, and potentially playing a key role in receptor functional modulation. Additionally, electrostatic and hydrophobic interactions further reinforced the compactness and structural complementarity at the binding interface. Fig 7e-Fig 7h present the key interactions of Compounds 2–5: Compound 2 forms a hydrogen bond with ASN-455; Compound 3 interacts with HIS-398, ARG-394, and GLY-442; Compound 4 establishes multiple hydrogen bonds with ASN-455, ARG-515, and THR-483; Compound 5 engages in hydrogen bonding with ALA-430. Overall, these high-activity compounds demonstrate stable multi-site binding with ERα, and their binding energy trends align with predicted activity values, providing structural-level validation of the predictive model.

## Conclusion

In this study, the DFFS method was employed to identify key two-dimensional molecular descriptors of ERα antagonists, and multiple ensemble learning algorithms were integrated to construct a two-stage predictive framework for both

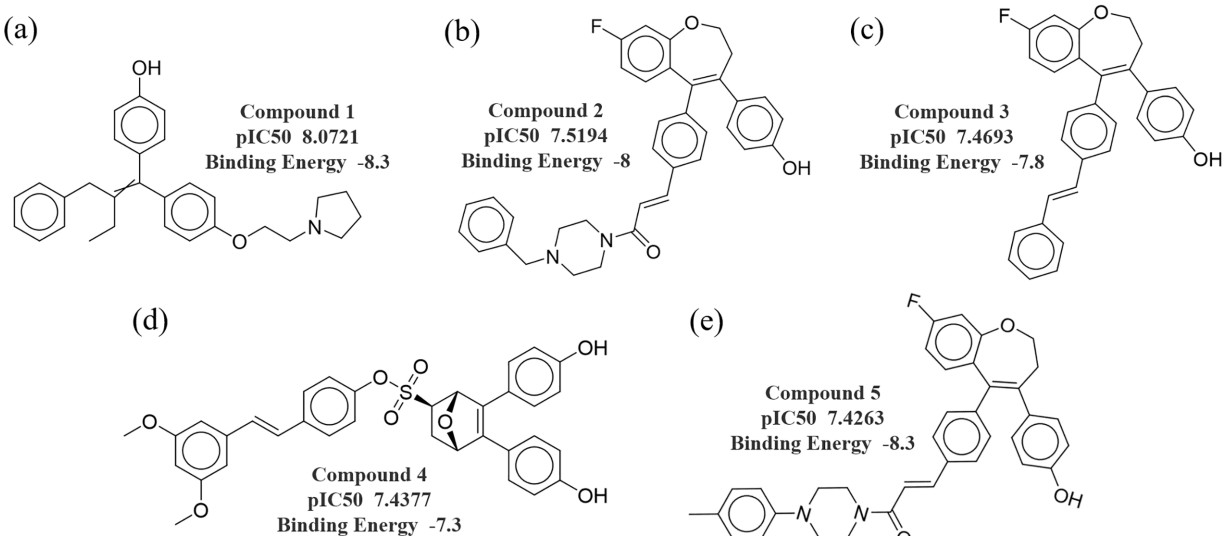

**Fig 6. Activity prediction and binding energy analysis of high-activity compounds.** (a)–(e) correspond to the predicted pIC50 values and molecular docking binding energies of the five high-activity compounds.

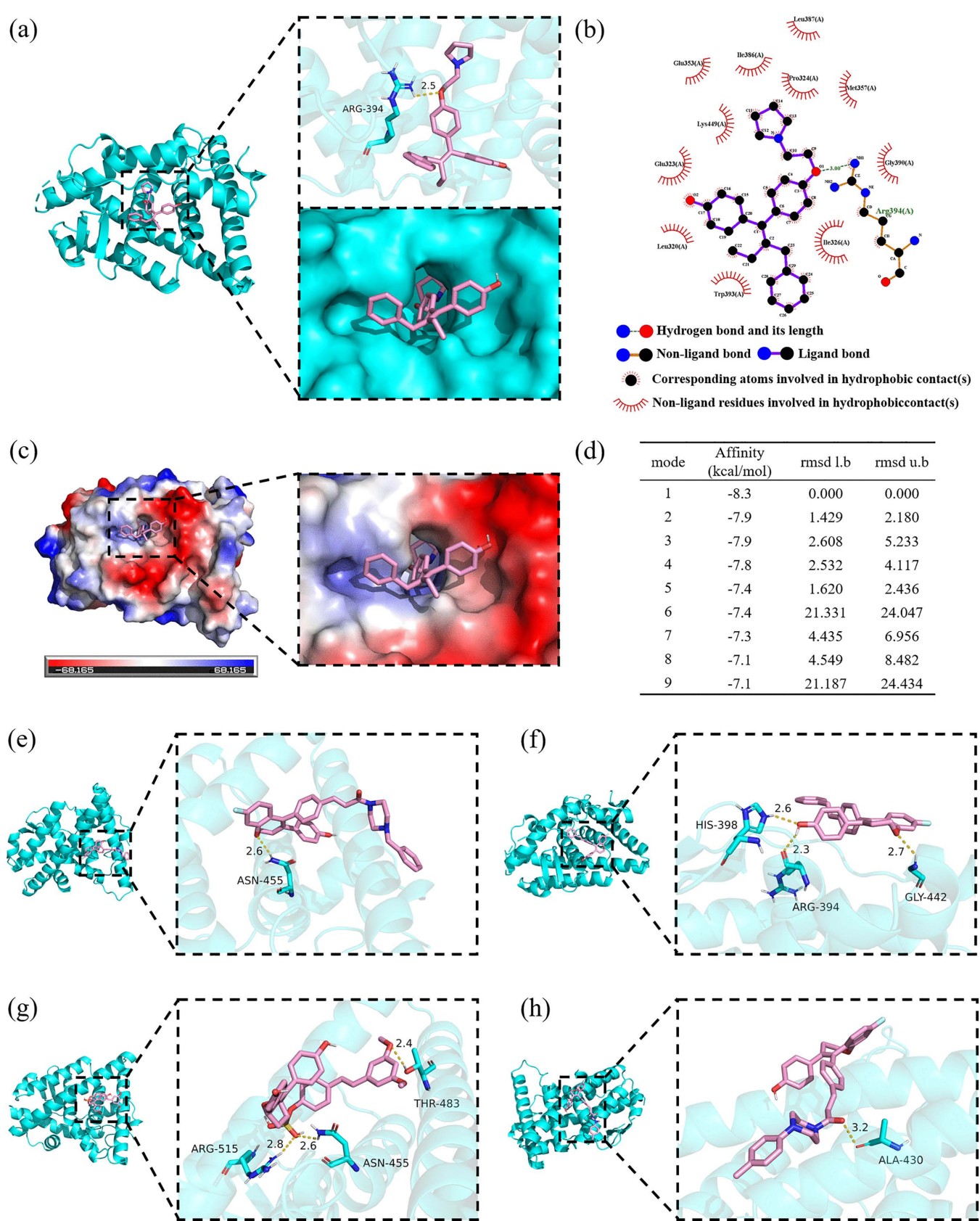

**Fig 7. Molecular docking and interaction analysis of high-activity compounds with ERα.** (a)–(d) show the three-dimensional binding confor-mation, two-dimensional interaction map, electrostatic potential distribution, and binding energy analysis of Compound 1; (e)–(h) display the three-dimensional binding conformations of Compounds 2–5 with ERα (hydrogen bonds are indicated by yellow dashed lines).

compound bioactivity and critical pharmacological properties. This approach enabled systematic screening and property evaluation of high-activity candidate compounds. The results demonstrated that LightGBM achieved the best perfor-mance in bioactivity prediction, while the Stacking model exhibited high reliability across various pharmacological property predictions. Further ablation studies analyzed the interrelationships among pharmacological properties of high-activity compounds using the Phi coefficient and employed SHAP analysis to identify core molecular features, namely Lipoaf-finityIndex and MDEC-23, as major contributors to compound activity. Finally, molecular docking analyses—including three-dimensional binding conformations, two-dimensional interaction maps, electrostatic potential distributions, and binding energy calculations—validated the binding affinity of high-activity compounds with the target protein, providing quantitative support for the predictive model.

Although this study has achieved notable progress, several limitations remain to be addressed in future work. First, a certain degree of class imbalance was observed in the ADMET binary classification datasets, though the disparity was rel-atively moderate. Given that model training in this study prioritized data authenticity and representativeness, oversampling and undersampling techniques were not applied. We also attempted to employ SMOTE to mitigate potential bias intro-duced by class imbalance; however, the improvement in predictive performance was marginal. Second, in terms of feature construction, this study utilized only two-dimensional molecular descriptors. While these descriptors offer high computa-tional efficiency and strong interpretability, they cannot directly capture the three-dimensional spatial conformations, elec-tronic density distributions, and stereochemical features of molecules—factors that play a critical role in target recognition and binding energy estimation. Finally, in the molecular validation stage, although molecular docking analyses confirmed the rationality of the predicted results, molecular dynamics (MD) simulations were not yet performed. MD simulations can dynamically evaluate conformational fluctuations, hydrogen-bond stability, and free energy variations of ligand–receptor complexes, providing a more comprehensive understanding of intermolecular interaction mechanisms. Future research will integrate three-dimensional structural quantitative descriptors and molecular dynamics simulations, while expanding dataset diversity and optimizing sample balance strategies, to further enhance the model's generalization ability, predictive accuracy, and biological interpretability.

## Acknowledgments

The authors are grateful to the editors and reviewers for their suggestions, which have improved this paper.

## Author contributions

**Data curation:** Jinhui Cao.

**Formal analysis:** Jinhui Cao.

**Methodology:** Yanli Liu.

**Software:** Jinhui Cao.

**Supervision:** Yanli Liu.

**Visualization:** Jinhui Cao.

**Writing – original draft:** Jinhui Cao.

**Writing – review & editing:** Yanli Liu.

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
