## [Decision Letter · Decision Letter 0]

13 Sep 2025

Dear Dr. Liu,

Thank you for submitting your manuscript to PLOS ONE. After careful consideration, we feel that it has merit but does not fully meet PLOS ONE’s publication criteria as it currently stands. Therefore, we invite you to submit a revised version of the manuscript that addresses the points raised during the review process.

**ACADEMIC EDITOR:**
**In case any additional references are suggested during the peer-review process, they should only be included if the authors agree that they are relevant and useful.**

We look forward to receiving your revised manuscript.

Kind regards,

Khalid Raza, PhD (Computational Biology)

Academic Editor

PLOS ONE

Journal Requirements:

Reviewers' comments:

Reviewer's Responses to Questions

**Comments to the Author**

1. Is the manuscript technically sound, and do the data support the conclusions?

Reviewer #1: Yes

Reviewer #2: Partly

2. Has the statistical analysis been performed appropriately and rigorously?

Reviewer #1: Yes

Reviewer #2: N/A

3. Have the authors made all data underlying the findings in their manuscript fully available?

Reviewer #1: Yes

Reviewer #2: Yes

4. Is the manuscript presented in an intelligible fashion and written in standard English?

Reviewer #1: No

Reviewer #2: Yes

Reviewer #1: The authors preparing a well-structured and technically solid manuscript. The study addresses an important problem in drug design by proposing a two-stage QSAR framework combined with SHAP analysis, which adds both methodological rigor and interpretability. The integration of feature selection, ensemble learning, and explainability is timely and relevant. I found the manuscript generally clear and logically organized. That said, I have several comments and suggestions that, if addressed, would further improve the clarity, robustness, and impact of the work:

Comment 1. The manuscript mentions 1,974 compounds retrieved from ChEMBL, but it is unclear whether activity values were standardized across assay types (e.g., binding affinity vs. functional assays). Please clarify the inclusion/exclusion criteria for compounds and how assay heterogeneity was handled, as inconsistent experimental data can bias QSAR models.

Comment 2. High AUC scores are reported, but no independent external validation set is used. This raises concerns about model generalizability beyond the training distribution.

Comment 3. The Phi coefficient correlation analysis among ADMET properties is interesting, but the discussion risks over-interpretation without biological literature support.

Comment 4. The threshold of pIC50 > 6 for active compounds should be pharmacologically justified, as the choice may influence screening outcomes.

Comment 5. The discussion of limitations is rather brief. Points such as imbalanced ADMET datasets, exclusion of 3D descriptors, and the absence of docking/molecular dynamics validation should be more explicitly acknowledged to provide transparency for future improvements.

Comment 6. SHAP analysis identifies key descriptors (LipoaffinityIndex, MDEC-23), but the mechanistic links to ERα antagonism need stronger justification with case-based examples.

Comment 7. The study lacks benchmarking against recent state-of-the-art QSAR/ML or ChemBERTa-based models for ERα prediction, limiting claims of novelty.

Comment 8. The Dual-Filter Feature Selection (DFFS) method is novel, but its stability should be validated across cross-validation folds or bootstrap sampling to ensure reproducibility of selected descriptors.

Reviewer #2: The manuscript titled "Multi-objective QSAR Prediction of ERα Antagonists via SHAP-based Interpretation" proposes a two-stage predictive framework combining QSAR and machine learning to evaluate both the biological activity and ADMET properties of drug candidates. While the study provides interesting results, several aspects require improvement, including:

Please see attached file

**Do you want your identity to be public for this peer review?** For information about this choice, including consent withdrawal, please see our Privacy Policy

Reviewer #1: **Yes: ** Nagmi Bano

Reviewer #2: No

---

## [Author Response · Author response to Decision Letter 1]

5 Nov 2025

We have completed the revisions to the paper and have re-uploaded the manuscript, figures, response letter, and other materials.

---

## [Decision Letter · Decision Letter 1]

18 Nov 2025

Multi-objective QSAR Prediction of ERα Antagonists via SHAP-based Interpretation

PONE-D-25-46112R1

Dear Dr. Liu,

We’re pleased to inform you that your manuscript has been judged scientifically suitable for publication and will be formally accepted for publication once it meets all outstanding technical requirements.

Kind regards,

Khalid Raza, PhD (Computational Biology)

Academic Editor

PLOS ONE

Additional Editor Comments (optional):

I am pleased to inform you that your paper has been accepted for publication in the current form. Following a rigorous peer review process, your manuscript received positive feedback from the reviewers and the editorial team. Your research offers a valuable contribution to the field, and we are confident that it will be of significant interest to our readership. On behalf of the editorial board, I extend our warmest congratulations.

Reviewers' comments:

Reviewer's Responses to Questions

**Comments to the Author**

Reviewer #1: (No Response)

2. Is the manuscript technically sound, and do the data support the conclusions?

Reviewer #1: Yes

3. Has the statistical analysis been performed appropriately and rigorously?

Reviewer #1: Yes

4. Have the authors made all data underlying the findings in their manuscript fully available?

Reviewer #1: Yes

5. Is the manuscript presented in an intelligible fashion and written in standard English?

Reviewer #1: Yes

Reviewer #1: The authors have thoroughly addressed all comments from the previous review round. The manuscript has improved significantly in clarity, methodological justification, and overall scientific rigor. All concerns regarding methodology, interpretation, and presentation have been satisfactorily resolved. I find no remaining issues that require further revision. The manuscript is now suitable for publication, and I recommend acceptance in its current form.

**Do you want your identity to be public for this peer review?** For information about this choice, including consent withdrawal, please see our Privacy Policy

Reviewer #1: No

---

## [Editor Report · Acceptance letter]

PONE-D-25-46112R1

PLOS One

Dear Dr. Liu,

I'm pleased to inform you that your manuscript has been deemed suitable for publication in PLOS One. Congratulations! Your manuscript is now being handed over to our production team.

Kind regards,

on behalf of

Dr. Khalid Raza

Academic Editor

PLOS One